

# Activity of biogenic silver nanoparticles in planktonic and biofilm-associated *Corynebacterium pseudotuberculosis*

Laerte Marlon Santos[1], Daniela Méria Rodrigues[1], Bianca Vilas Boas Alves[1], Mauricio Alcântara Kalil[1], Vasco Azevedo[2], Debmalya Barh[2,3], Roberto Meyer[1], Nelson Duran[4], Ljubica Tasic[5] and Ricardo Wagner Portela[1]

[1] Instituto de Ciencias da Saude, Universidade Federal da Bahia, Salvador, Bahia, Brazil
[2] Instituto de Ciencias Biologicas, Universidade Federal de Minas Gerais, Belo Horizonte, Minas Gerais, Brazil
[3] Institute of Integrative Omics and Applied Biotechnology, Nonakuri, West Bengal, India
[4] Instituto de Biologia, Universidade Estadual de Campinas, Campinas, Sao Paulo, Brazil
[5] Instituto de Quimica, Universidade Estadual de Campinas, Campinas, Sao Paulo, Brazil

Corresponding author
Ricardo Wagner Portela,
rwportela@ufba.br

## ABSTRACT

*Corynebacterium pseudotuberculosis* is a gram-positive bacterium and is the etiologic agent of caseous lymphadenitis (CL) in small ruminants. This disease is characterized by the development of encapsulated granulomas in visceral and superficial lymph nodes, and its clinical treatment is refractory to antibiotic therapy. An important virulence factor of the *Corynebacterium* genus is the ability to produce biofilm; however, little is known about the characteristics of the biofilm produced by *C. pseudotuberculosis* and its resistance to antimicrobials. Silver nanoparticles (AgNPs) are considered as promising antimicrobial agents, and are known to have several advantages, such as a broad-spectrum activity, low resistance induction potential, and antibiofilm activity. Therefore, we evaluate herein the activity of AgNPs in *C. pseudotuberculosis*, through the determination of minimum inhibitory concentration (MIC), minimum bactericidal concentration (MBC), antibiofilm activity, and visualization of AgNP-treated and AgNP-untreated biofilm through scanning electron microscopy. The AgNPs were able to completely inhibit bacterial growth and inactivate *C. pseudotuberculosis* at concentrations ranging from 0.08 to 0.312 mg/mL. The AgNPs reduced the formation of biofilm in reference strains and clinical isolates of *C. pseudotuberculosis*, with interference values greater than 80% at a concentration of 4 mg/mL, controlling the change between the planktonic and biofilm-associated forms, and preventing fixation and colonization. Scanning electron microscopy images showed a significant disruptive activity of AgNP on the consolidated biofilms. The results of this study demonstrate the potential of AgNPs as an effective therapeutic agent against CL.

## INTRODUCTION

*Corynebacterium pseudotuberculosis* is a gram-positive pathogen that has great veterinary importance since it is the etiologic agent of caseous lymphadenitis (CL) in small ruminants

(*Dorella et al., 2006*). CL is mainly characterized by the development of encapsulated granulomas in visceral and superficial lymph nodes, as well as in organs such as liver, lungs, spleen, and kidneys (*Barral et al., 2022*). As *C. pseudotuberculosis* is a zoonotic agent, it is important to consider CL as a public health problem, since cases of infection in humans were already described in several countries, such as Australia, New Zealand, and France (*Bastos et al., 2012*).

CL is refractory to antibiotic therapy because of the development of a thick capsule around the lesions, and as a consequence of the cheesy nature of the granuloma's content. Also, antibiotics can only eliminate the pathogen from superficial lesions, and internal lesions and/or abscesses may not be affected by the treatment and remain active, causing recurrences (*Williamson, 2001*; *Fontaine & Baird, 2008*).

The ability of the bacteria from *Corynebacterium* genus to produce biofilm is considered as an important virulence factor (*Guedes et al., 2015*). This structure is an aggregate of microcolonies surrounded by a matrix composed by polysaccharides, forming organized communities that allow adhesion to biological surfaces, and is characterized by an enhanced resistance to antimicrobials (*Vestby et al., 2020*) and biocides (*Sá et al., 2013*) in the chronic stage of the disease.

Currently, there is a need to develop new alternatives for the treatment and control of infectious diseases, and silver nanoparticles (AgNPs) are being considered as promising antimicrobial agents, since it exhibits several advantages, such as broad-spectrum and wound healing activities, and a low potential of resistance induction (*Tăbăran et al., 2020*; *Fernandez et al., 2021*). AgNPs produced using *Fusarium oxysporum* biomass have already proven to have significant antimicrobial activity against *Candida* sp., not only *in vitro* (*Fonseca et al., 2022*) but also *in vivo* (*Fonseca et al., 2023*). A previous study found that biogenic AgNPs were able to accelerate the wound healing process after the surgical excision of caseous lymphadenitis lesions (*Santos et al., 2019*). Therefore, we aimed herein to evaluate the antibiofilm and antibacterial activity of AgNPs in *C. pseudotuberculosis* clinical isolates and reference strains.

## MATERIALS & METHODS

### Biogenic AgNP synthesis and characterization

The AgNPs were synthetized as described by *Ballottin et al. (2017)*. The *Fusarium oxysporum* fungus was cultivated in a solid culture medium (2% malt, 2% agar, 0.5% yeast extract and distilled water), and was incubated for one week at 28 °C. Then, sterile milliQ water was added to the culture until a 0.1 g/mL protein concentration was reached, and kept under agitation for 72 h. Then, the supernatant was the filtered and 0.01 mol/L of $AgNO_3$ was added. The solution was sealed with aluminum foil and kept at 28 °C until the formation of the nanoparticles.

The morphology of the AgNPs was observed using a Zeiss CEM-902 (Zeis, Oberkochen, Germany) transmission electron microscope set at 80 keV. The zeta potential of the AgNPs was obtained by electrophoretic mobility, through the dispersion of the AgNP in a KCl solution. The surface charge was measured using a Zetasizer Nano series equipment (Malvern Instruments, Malvern, UK) (*Ballottin et al., 2017*)

## Bacterial strains

Four *C. pseudotuberculosis* reference strains were used in this study: the 1002 strain, which was used as a reference for the bacterial genome project (*Mariano et al., 2016*); the N1 strain, which is a viscerotropic strain isolated from a lung lesion of a sheep with CL (*Loureiro et al., 2016*); the T1 strain, an attenuated strain used as a vaccine model (*Moura-Costa et al., 2008*); and the CAPJ4 strain, a strain previously described as biofilm-producing (*De Sá et al., 2021*). All these strains belong to the biovar ovis and have already their genome sequenced and deposited (GenBank accession numbers CP001809.2, CP013146, CP015100.1 and CP026499, respectively). Four clinical isolates obtained from caseous samples of sheep subjected to CL lesion excision (*Kalil et al., 2019*) were molecularly identified using quadruplex PCR (*Almeida et al., 2017*), and were also included in this study.

## Determination of minimal inhibitory concentration (MIC) and minimal bactericidal concentration (MBC)

The broth microdilution methodology was performed as previously described by *Norman et al. (2014)*, with modifications. The solution containing the AgNPs was diluted in sterile milli-Q (concentrations ranging from 0.01 to 5 mg/mL). The *C. pseudotuberculosis* strains were then inoculated in brain heart infusion broth—BHI (HIMEDIA, Mumbai, India) added with 0.1% Tween 80 and incubated for 24 h. After this step, the strains were diluted in $2 \times$ BHI until an optical density of 0.08–0.10 at 600 nm was achieved (which contains approximately $2 \times 10^6$ CFU/mL of the bacteria). These suspensions were then diluted in $2\times$ BHI broth (concentration of $1 \times 10^6$ CFU/mL). After dilution, 100 μL of the AgNP colloidal solution in different concentrations and 100 μL of the inoculum were inoculated into a 96-well flat-bottom polystyrene microplate. The culture microplates were consequently incubated at 37 °C for 48 h. As controls, the AgNP-not treated *C. pseudotuberculosis* bacterial suspension was used as a positive control, and the colloidal solution of AgNPs without the addition of bacteria was considered as a negative control. After a final 600 nm read at a spectrophotometer (Bio-Rad, Hercules, CA, USA), it was possible to determine the minimum concentration of AgNP that was able to fully inhibit the bacterial growth ($MIC_{100}$). In addition, Petri dishes containing BHI agar were inoculated with 20 μL taken from each well of the culture microplates used for the $MIC_{100}$ determination, followed by an incubation for 48 h at 37 °C, and then the minimal concentration of the AgNPs that was able to fully inactivate the bacteria ($MBC_{100}$) was established. The $MIC_{100}$ and the $MBC_{100}$ assays were repeated three times.

## Biofilm production assay

This semiquantitative methodology was conducted as previously described by *Kalil et al. (2019)*. A *C. pseudotuberculosis* bacterial suspension was inoculated in tryptone soy broth (TSB) and incubated at 37 °C. 200 μL of the bacterial suspension were transferred to culture microplates and incubated for 48 h at 37 °C. Then, the wells were aspirated and washed twice with sterile PBS. The material that remained attached to the plate was then fixed with methanol and dried. The biofilms were stained with a 2% crystal violet solution
for 5 min and washed with 0.01 M PBS pH 7.2. The dye was then eluted with a 33% acetic acid solution. As a negative control, it was used the content of wells with the broth and without the inoculum. The 595 nm OD from each well was then measured.

The following equations were used to characterize the intensity of biofilm formation, where ODI indicates the optical density of the isolate, and ODNC represents the optical density of the negative control: ODI ≤ ODNC = no biofilm development; ODI/ODNC ≤ 2 = weak biofilm formation; ODI/ODNC ≤ 4 = moderate biofilm production capacity; ODI/ODNC >4 = strong biofilm production capacity (*Nostro et al., 2007*). The CAPJ4 strain of *C. pseudotuberculosis* was used as a biofilm-forming control strain (*Sá et al., 2018*). Three independent experiments were performed.

## Determination of minimal inhibitory concentration for consolidated biofilms (MIBC)

The AgNPs antibiofilm action assay was conducted as previously described by *Santos et al. (2021)*. The *C. pseudotuberculosis* isolates were inoculated in TSB and incubated at 37 °C for 48 h. The bacterial suspensions were then standardized to a 595 nm OD of 0.2, and 200 µL were transferred to culture microplate wells and incubated for 48 h at 37 °C. After the consolidation of the biofilm, and taking into consideration the optimal concentrations previously obtained in the minimal inhibitory concentration (MIC) assays, the bacterial inoculum used for biofilm formation, and the fact that concentrations of up to 16x MIC were previously described as needed to observe antibiofilm action (*De Oliveira et al., 2016*; *Trevisan et al., 2018*; *Shrestha et al., 2022*), 200 uL of the AgNP solutions (0.25, 0.5, 1, 2 and 4 mg/mL) were added to the wells. The microplates were then incubated for 48 h at 37 °C. The 595 nm OD was determined right after the addition of the AgNP (OD 0 h), 24 h (OD 24 h), and 48 h (OD 48 h) after. The MIBC was then characterized as the lowest antimicrobial concentration in which there was no time-dependent increase in the biofilm content when an early exposure time was compared with a later exposure time (*Macià, Rojo-Molinero & Oliver, 2014*). Different controls were included in the experiment: a negative control, composed by biofilm and Milli-Q water; a control with TSB broth and each AgNP dilution; and a control made only with TSB broth. Three independent experiments were performed.

## Determination of AgNP interference on biofilm formation

*C. pseudotuberculosis* isolates were inoculated in TSB and incubated at 37 °C for 48 h. The bacterial suspensions were then standardized to a 595 nm OD of 0.2. 100 µL of the AgNP solutions (0.25, 0.5, 1, 2 and 4 mg/mL) were mixed with 100 µL of these standardized bacterial suspensions and added to culture microplates, and incubated at for 48 h for 37 °C. After 48 h, the biofilm was detected and the percentage of inhibition of biofilm production was calculated considering the control bacterial suspensions that were not incubated with the AgNPs (*Kalil et al., 2019*). Three independent trials were performed.

The percentage of inhibition of the biofilm formation was calculated using a formula previously described (*Siddique et al., 2020*; *Santos et al., 2021*), as follows:

$$\% \text{ inhibition} = 1 - \frac{\text{OD595 of the treated } C.pseudotuberculosis}{\text{OD595 of the not} - \text{treated } C.pseudotuberculosis} \times 100.$$

### Scanning electron microscopy (SEM)

The bacterial biofilms (AgNP-treated and -untreated) were obtained using the methodologies described herein; however, a sterile glass coverslip was added to each well of the culture plate, and the coverslips containing the biofilms were then analyzed at the scanning electron microscope. The biofilms (treated or not with the AgNPs at 4 mg/mL) were fixed in (i) 2.5% glutaraldehyde in 0.1 M sodium cacodylate pH 7.4 for 2 h, and (ii) 1% osmium tetroxide in 0.1 M sodium cacodylate for 1 h. After the fixation step, the biofilms were dehydrated in ethanol (30, 50, 70, 90%, and absolute alcohol) and dried. The biofilms were then examined using the SEM JSM-6390LV scanning electron microscope (Jeol, Tokyo, Japan) operated at 15 kV.

### Statistical analysis

The statistical analyses were made using the SPSS v. 22.0 (IBM, Amonk, NY, USA) software. The distributions of the results of the broth microdilution, $MBC_{100}$ determination, biofilm formation and interference assays were verified using the D'Agostino–Pearson test, and the comparisons of the AgNP-treated and AgNP-not treated bacteria and of the different AgNPs concentration results were made using the $t$-test and the one-way ANOVA ($p < 0.05$).

## RESULTS

### Characterization of the AgNPs

The AgNPs used in this study were spherical, and presented a zeta potential of $-31.7 \pm 2.8$ mV. They were also characterized by a 0.231 polydispersity, and sizes of $28.0 \pm 13.1$ nm. These results confirmed what was previously described for these AgNPs (*Stanisic et al., 2018*).

### Detection of biofilm formation by *C. pseudotuberculosis* strains

A total of eight *C. pseudotuberculosis* isolates that were used in this study were classified as biofilm formers. The $OD_{595}$ of the negative control was 0.136, and $OD_{595}$ values between 0.136 and 0.272 determined the strains that were poor biofilm producers. If the $OD_{595}$ was between 0.272 and 0.544, the *C. pseudotuberculosis* strain was considered as an isolate with moderate biofilm production activity, and $OD_{595} > 0.544$ characterized the strain as a strong biofilm producer (*Nostro et al., 2007*). In this experiment, strains 1002 and T1 were characterized as poor biofilm producers. Strains N1 and CAPJ4, and clinical isolates 06 and 96, were characterized as moderate biofilm producers, while the clinical isolates 05 and 15 were considered as strong biofilm producers (Fig. 1).

### Antimicrobial susceptibility assessment of planktonic and biofilm cells of *C. pseudotuberculosis*

The evaluation of the antimicrobial activity of AgNP against *C. pseudotuberculosis* using the broth microdilution methodology demonstrated a growth inhibition of most isolates at a concentration of 0.156 mg/mL, with the lowest $MIC_{100}$ being 0.08 mg/mL. Similar

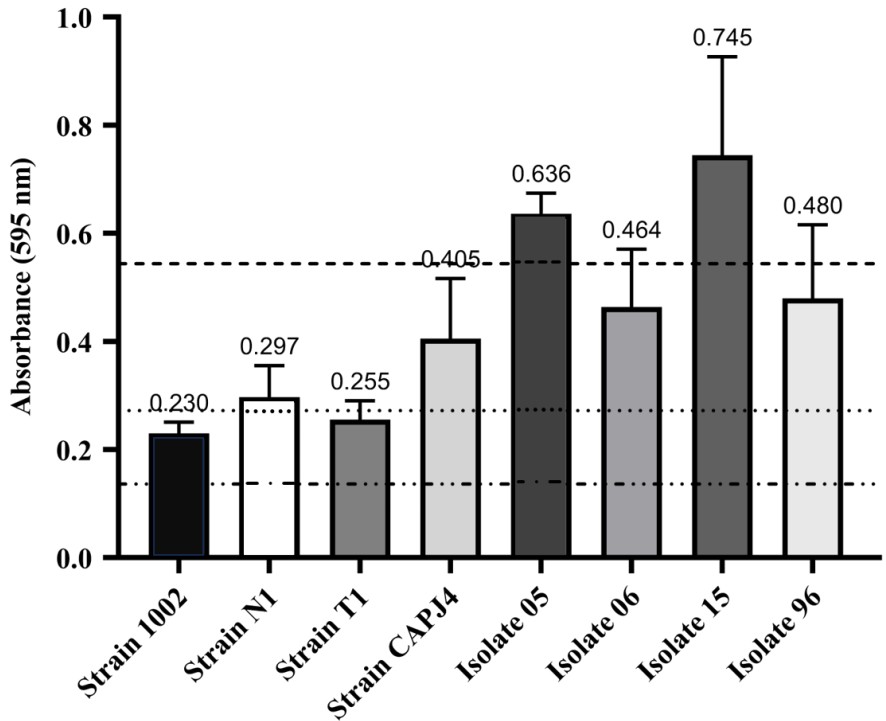

**Figure 1** **Biofilm formation by *Corynebacterium pseudotuberculosis* reference strains and clinical isolates.** Bars indicate the standard deviations. The dashed lines indicate the mean value of the negative control ($OD_{595}$ = 0.136), weak biofilm production ($OD_{595}$ between 0.136 and 0.272), moderate biofilm production ($OD_{595}$ between 0.272 and 0.544) and strong biofilm production ($OD_{595}$ > 0.544). The results express the means of three independent experiments.

results were observed among the reference strains for the bactericidal action of AgNPs, with $MBC_{100}$ values equivalent to 0.156 mg/mL. AgNPs had a lower bactericidal action among clinical isolates (Table 1). A $MIC_{100}$ curve was created to visualize the antibacterial action of the AgNPs at different concentrations (Fig. S1).

The evaluation of the action of the AgNPs in consolidated biofilms was measured at three incubation periods: 0 h, 24 h and 48 h. After adding the AgNP colloidal solution to the consolidated biofilms, all bacteria showed biofilm reductions at 24 h and 48 h of incubation and at the different tested concentrations. The greatest decreases were observed for the consolidated biofilms of the clinical isolates, except for isolate 06, and the CAPJ4 strain. The consolidated biofilms from strains 1002, N1, and T1 were less sensitive to the action of AgNPs (Fig. 2).

The interference on biofilm formation assay showed that, within 48 h of incubation, the AgNPs were able to significantly interfere with the formation of biofilm by the bacterial strains, with interference values greater than 80% at a AgNP concentration of 4 mg/mL for the reference strains N1 and CAPJ4. The percentages of interference in biofilm formation by the AgNPs are shown in Table 2. The AgNPs were able to prevent biofilm formation

**Table 1** Sensitivity profile of *C. pseudotuberculosis* reference strains and clinical isolates to AgNPs.

| Strains of *C. pseudotuberculosis* | MIC$_{100}$ (mg/mL) | MBC$_{100}$ (mg/mL) |
| --- | --- | --- |
| 1002 | 0.156 | 0.156 |
| N1 | 0.156 | 0.156 |
| T1 | 0.156 | 0.156 |
| CAPJ4 | 0.08 | 0.156 |
| Isolate 05 | 0.312 | 0.312 |
| Isolate 06 | 0.156 | 0.312 |
| Isolate 15 | 0.08 | 0.312 |
| Isolate 96 | 0.156 | 0.312 |

Notes.

The results express the means of three different experiments.

MIC$_{100}$, minimum AgNP concentration capable to inhibit 100% of the bacterial growth; MBC$_{100}$, minimum AgNP concentration capable to kill all the bacteria.

by the reference strains at all concentrations tested. Also, a significant reduction in biofilm formation was observed in all clinical isolates at the concentrations tested herein (Fig. 3).

## Scanning electron microscopy

SEM micrographs obtained from AgNP-treated and AgNP-not treated *C. pseudotuberculosis* are shown in Figs. 4A and 4B. Untreated *C. pseudotuberculosis* was typically presented as cocci or coccobacilli, with a smooth, intact cell wall, covered by an amorphous exopolysaccharide matrix.

After the exposure to the AgNP colloidal solution (4 mg/mL) for 48 h, the amount of *C. pseudotuberculosis* biofilm significantly reduced (Fig. 4C), with a large quantity of debris and materials deposited on the bacterial surface (Fig. 4D), and even a complete disruption of the bacterial biofilm, with apparent loss of adhesion (Fig. 4E). Some bacteria that remained intact presented surface vesicles (Fig. 4F), indicating an increased plasma membrane permeability and the release of intracellular components.

## DISCUSSION

*C. pseudotuberculosis* is the etiological agent of CL, a disease that causes a significant reduction in the productivity and in the reproductive efficiency of infected animals, leading to economic losses (*Costa et al., 2022*). In addition, the treatment of the disease is generally refractory to antibiotic therapy. With the objective to develop new therapeutical options for CL, we identified that *Fusarium oxysporum*-based AgNPs can be a promising antimicrobial agent, because of its marked antibacterial action on *C. pseudotuberculosis* planktonic cells and on its associated biofilm.

Semi-quantitative analysis are valid tools to determine the capacity of biofilm formation, and it has been described that many bacterial isolates do not produce or are poor biofilm producers, and this fact is correlated with different sensitivities to antimicrobial agents (*Sá et al., 2013*). Considering this situation, we included in our study different *C. pseudotuberculosis* reference strains and clinical isolates, which showed distinct biofilm production capacities. Interestingly, the CAPJ4 strain was recently used as a reference for

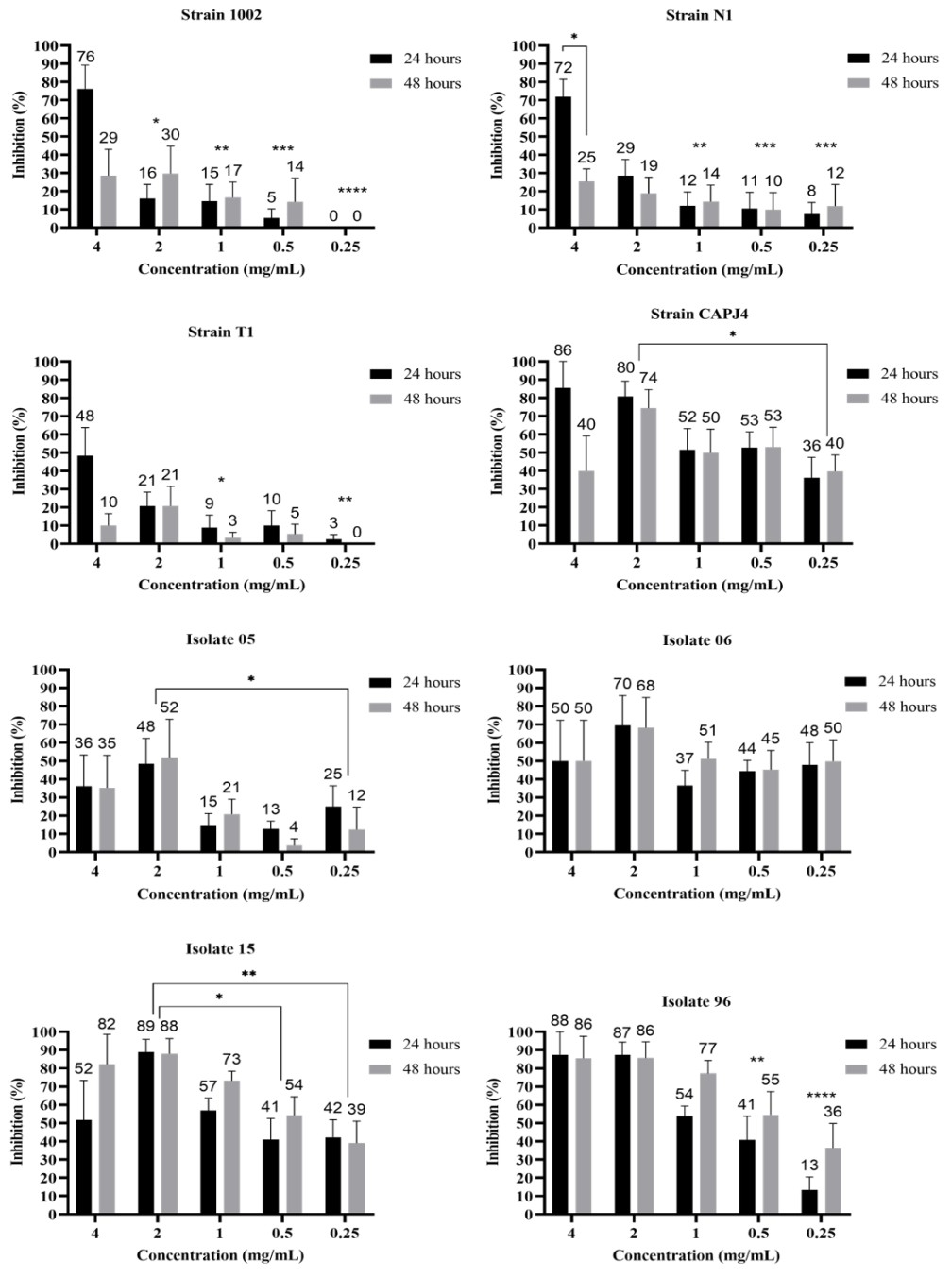

**Figure 2 AgNP activity on consolidated biofilms of *C. pseudotuberculosis* reference strains and clinical isolates after 24 and 48 hours of treatment.** Results express the inhibition means (in %) from three independent experiments. The bars represent the standard deviations. Asterisks indicate statistical differences between bacterial cells treated with different concentrations of AgNP and the not-treated control, at different times, analyzed using analysis of variance (one-way ANOVA) and the $t$-test. $*p < 0.03$, $**p < 0.005$, $***p < 0.0009$ and $****p < 0.0001$.

**Table 2  AgNP interference in *C. pseudotuberculosis* biofilm formation.** The results express the percentage of inhibition ± standard deviation (confidence interval of 95%) of biofilm formation after 48 h of incubation with AgNPs, as means of three independent experiments.

| Strain of *C. pseudotuberculosis* | AgNP (mg/mL) | | | | |
|---|---|---|---|---|---|
| | 4 | 2 | 1 | 0.5 | 0.25 |
| 1002 | 78 ± 19 (57–99) | 53 ±29 (22–84) | 42 ± 17 (23–60) | 68 ±20 (47–89) | 52 ± 23 (27–76) |
| N1 | 86 ± 16 (69–100) | 65 ± 24 (39–91) | 49 ± 17 (33–64) | 59 ± 17 (41–77) | 68 ± 12 (55–81) |
| T1 | 75 ± 34 (40–100) | 84 ± 8 (76–93) | 66 ± 16 (49–82) | 87 ±10 (77–97) | 62 ± 17 (44–80) |
| CAPJ4 | 91 ± 11 (80–100) | 86 ± 12 (73–99) | 83 ± 10 (73–94) | 90 ± 5 (85–95) | 89 ± 4 (84–93) |
| Isolate 05 | 59 ±22 (36–83) | 71 ± 14 (55–86) | 71 ± 8 (62–80) | 80 ± 9 (70–91) | 79 ± 9 (70–89) |
| Isolate 06 | 66 ± 29 (36–97) | 79 ± 6 (72–86) | 59 ± 20 (38–61) | 79 ± 11 (67–91) | 81 ± 8 (72–89) |
| Isolate 15 | 73 ±18 (54–93) | 79 ± 12 (63–89) | 77 ± 4 (73–81) | 89 ± 4 (85–93) | 88 ± 4 (83–92) |
| Isolate 96 | 78 ± 9 (68–87) | 83 ± 12 (7–95) | 83 ± 7 (75–90) | 88 ± 6 (82–94) | 87 ± 6 (81–93) |

**Notes.**

SD, Standard Deviation; CI 95%, confidence intervals.

a proteomic analysis focused on its virulence and capacity of biofilm production, and exhibited a marked up-regulation of the galactose-1-phosphate uridylyltransferase and N-acetylmuramoyl-L-alanine amidase enzymes, which are involved in biofilm formation and exopolysaccharide biosynthesis (*De Sá et al., 2021*).

Based on global metabolomic profiles, bacteria with weak and strong biofilm formation capacities present themselves as two distinct groups. Bacteria that are classified as poor biofilm producers can express more metabolites than bacteria with strong biofilm production; this situation indicates a higher endogenous metabolic activity (especially lipid metabolism), on which some isolates show a greater tendency to remain in their planktonic and free-floating state, than bacteria with strong biofilm production (*Wong et al., 2018*). Such results may justify the distinct profiles of biofilm production among *C. pseudotuberculosis* isolates. In addition, bacteria with weak to strong adherent biofilms can present multidrug resistant phenotypes and this characteristic can play a vital role in pathological processes based on the bacterial resistance to the host immunological system, and to antimicrobial agents (*Diriba et al., 2020*).

The results of the sensitivity test using the broth microdilution methodology demonstrated the effectiveness of AgNPs to inhibit the growth and in the inactivation of *C. pseudotuberculosis* reference strains and clinical isolates. This result agrees with studies showing the antibacterial activity of AgNPs obtained by green synthesis methods against *E. coli*, *Klebsiella pneumoniae*, *Salmonella* Typhimurium and *Salmonella* Enteritidis (*Loo et al., 2018*). *Punjabi et al. (2018)* concluded that AgNPs obtained by an extracellular synthesis methodology using *Pseudomonas hibiscicola* showed high efficiency against multi-drug-resistant (MDR) clinical isolates of methicillin-resistant *Staphylococcus aureus* (MRSA), extended spectrum β lactamases (ESBL) producer *K. pneumoniae*, vancomycin-resistant *Enterococcus faecalis* (VRE), and *M. tuberculosis*, with MICs ranging from 0.6–1.5 mg/mL; the results obtained by these authors showed the antimicrobial potential of AgNPs against resistant bacterial strains isolated from human hospitals. Bacterial inhibition tests also show that the effectiveness of AgNPs is dependent on size and shape, with spherical and

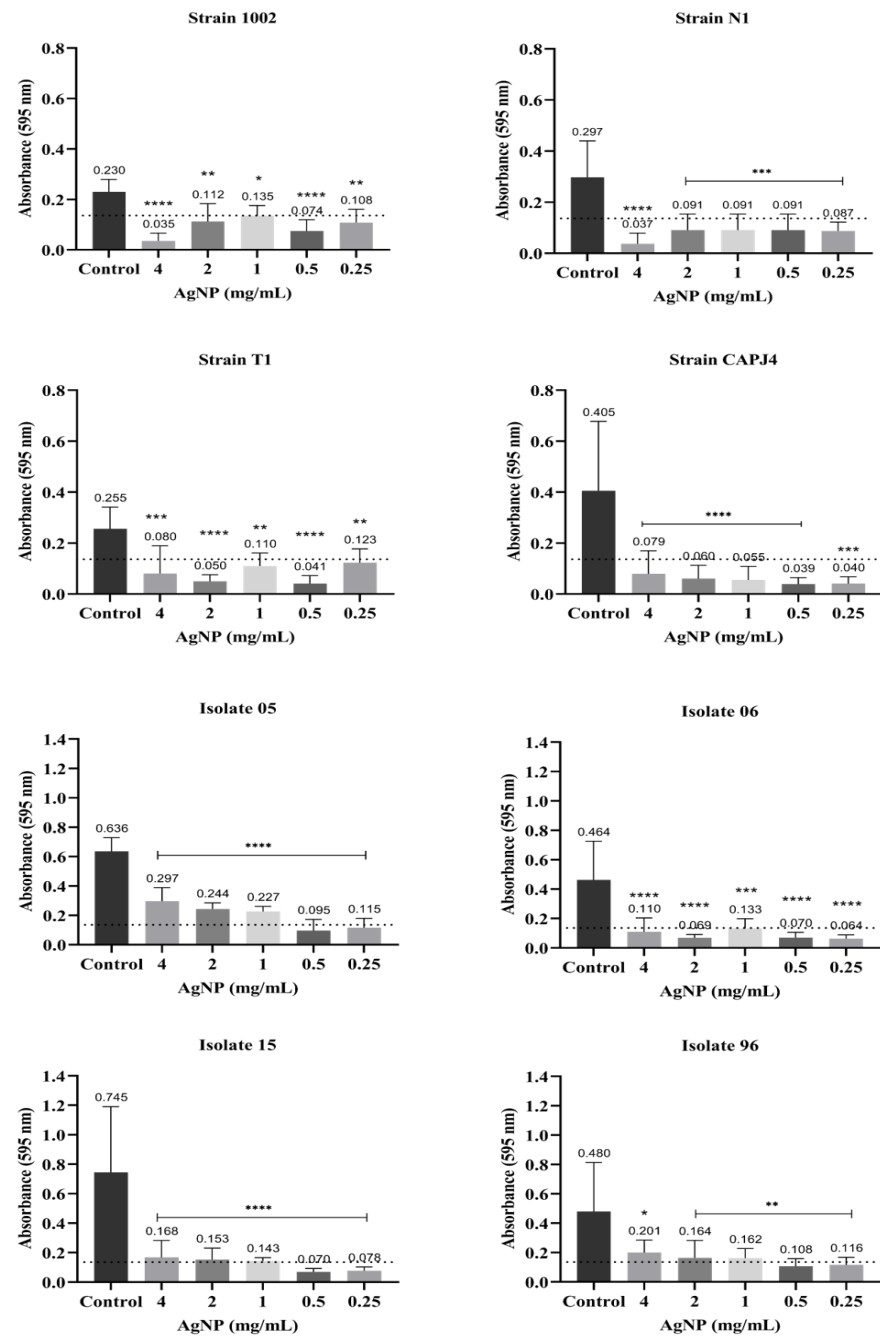

**Figure 3** **AgNP interference in the biofilm formation by *C. pseudotuberculosis* reference strains and clinical isolates.** Bars represent standard deviations. The dashed line indicates the mean value of the negative control ($OD_{595} = 0.136$). Asterisks indicate statistical differences between bacterial cells not treated and treated with different concentrations of nanoparticles, analyzed using analysis of variance (one-way ANOVA). *$p < 0.05$; **$p < 0.005$; ***$p < 0.0003$; ****$p < 0.0001$.

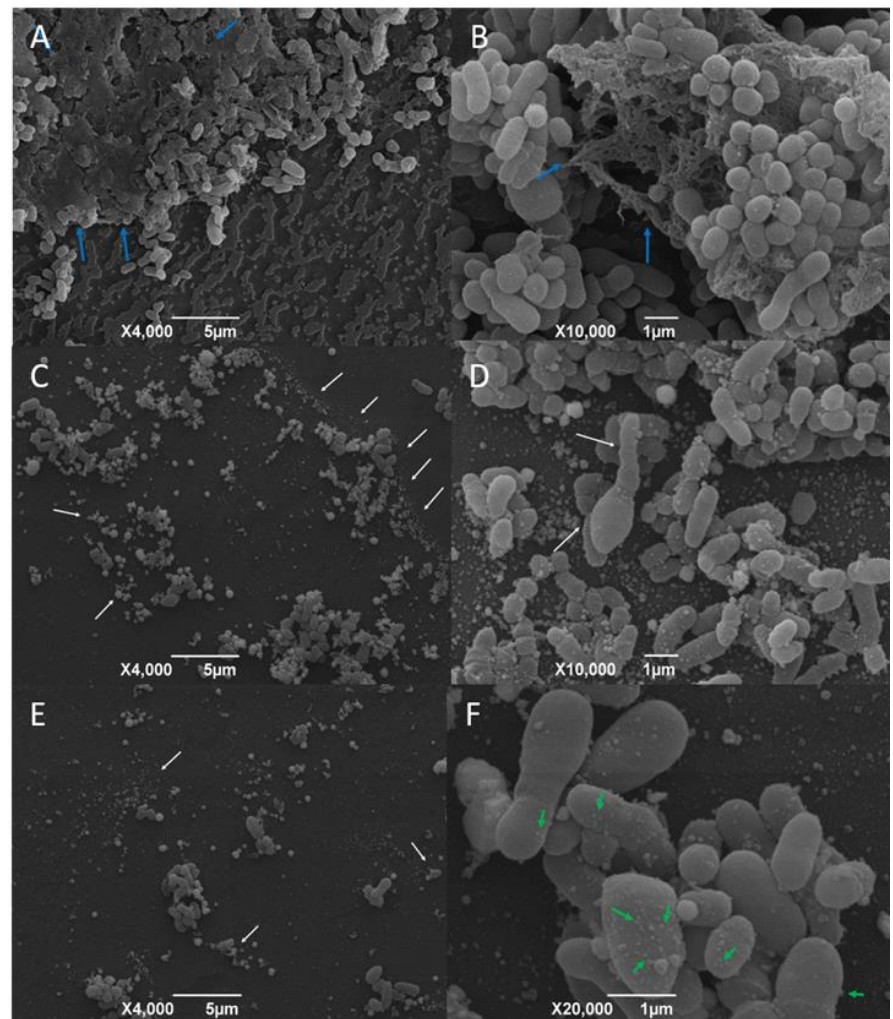

**Figure 4** **Representative scanning electron micrographs of the structure of mature *C. pseudotuberculosis* biofilms untreated (A and B) and treated with AgNPs at the concentration of 4 mg/mL (C, D, E and F).** Blue arrows indicate the presence of an exopolysaccharide matrix in the mature biofilm. White arrows indicate cell deformities and debris. Green arrows point to the presence of vesicles on the bacterial surface.

smaller AgNPs being more efficient (*Raza et al., 2016*), like the nanoparticles used against *C. pseudotuberculosis* herein.

Few studies report the action of antimicrobials on *C. pseudotuberculosis* biofilm. *Sá et al. (2013)* analyzed the effectiveness of disinfectants in interfering with *C. pseudotuberculosis* biofilm formation and in the consolidated biofilm, and they found that disinfectants were not effective against the consolidated biofilm; however, they observed that iodine inhibited the biofilm formation in 33% of the bacterial isolates, and quaternary ammonia was able to prevent the biofilm formation by 28% of isolates. It is noteworthy that the consolidation of a biofilm is a part of the attachment process, when there is a formed exopolymer that connects the bacteria between themselves and with the surface (*Garrett, Bhakoo &*

*Zhang, 2008*). Our study is the first one that evaluated the action of nanoparticles on *C. pseudotuberculosis* biofilm. However, the antibiofilm mechanisms of silver nanoparticles are not fully known, and can be dependent on the nanoparticle physicochemical properties and on the type of microorganism (*Durán et al., 2016*).

The results of the present study indicated that the consolidated biofilms of *C. pseudotuberculosis* suffered the action of AgNP at different concentrations. Natural compounds, such as extracellular fungal compounds, may be more effective stabilizing agents, since they prevent particle aggregation and enhance the antimicrobial activity of AgNPs, increasing bacterial inactivation in the biofilm and reducing nanoparticle cytotoxicity (*Liu et al., 2019*). The same AgNPs used in our study were administered as an ointment (with an AgNP concentration of 2.56 mg/mL) to sheep and goats with surgical wounds, and induced an accelerated wound healing process, in the absence of any clinical sign of toxicity and without any influence on hematological and clinical biochemistry parameters that could indicate a toxigenic process induced by the AgNPs (*Santos et al., 2019*). According to *Estevez et al. (2020)*, AgNPs produced with extracellular extracts of *Phanerochaete chrysosporium* cells (PchNPs) exhibited an activity based on the eradication of *E. coli* and *C. albicans* biofilms. *Siddique et al. (2020)* concluded that the percentage of biofilm inhibition by AgNPs in *K. pneumoniae* MDR strain MF953600 was 64%, and 84% for the MF953599 strain at the concentration of 100 mg/mL. The production of the exopolymer substance decreased after the AgNP treatment, while cellular protein leakage increased due to higher rates of cell membrane disruption. Size, shape, surface, and interior properties of nanoparticles (NPs) are important factors to be considered in biofilm control (*Liu et al., 2019*). The size of NPs is a crucial factor to be considered regarding its penetration into biofilms and should not exceed the dimensions of water channels in these biofilms. As the sizes of water channels in biofilms are difficult to estimate, the ideal size of NPs that can present a significant activity in biofilms should range between 5 and 100–200 nm (*Peulen & Wilkinson, 2011*), and the AgNPs used in this study presented sizes of $28.0 \pm 13.1$ nm. On the other hand, the reduced sensitivity of some consolidated *C. pseudotuberculosis* biofilms may be related to a lower diffusion and adsorption of the antimicrobial into the exopolymer matrix, which only allows the transport of nutrients and residual metabolites through the water channels and hinders the transport of antimicrobial agents. Considering this situation, only the bacteria present on the surface of the biofilm can be affected (*Koo et al., 2017*).

The difference in the action of AgNPs in biofilms of reference strains and clinical isolates can be attributed to the constant maintenance of the reference strains in the culture media used in the experiments. *Djais et al. (2019)*, who evaluated the effect of propolis extract in the formation of biofilm by *Streptococcus mutans*, describe that this situation can be a consequence of the sucrose and other polysaccharides present in the TSB medium. Sucrose is used by oral streptococci to produce extracellular polysaccharides present in dental biofilms. Glucans are essential for dental plaque formation and increase the attachment of bacteria to the tooth surface, and it is known that fructan can enhance the biofilm virulence, since it is a binding site for *S. mutans* adhesion (*Djais et al., 2019*).

The biofilm formation interference test is one of the strategies adopted to control the stages of biofilm development. The objective of the interference test is to inhibit the initial fixation of bacteria on biofilm-forming surfaces, thus reducing the chances of biofilm development (*Subhadra et al., 2018*). Biofilm formation control strategies have been widely used in the biomedical field, such as the use of antibiotic-coated catheters (*Balne et al., 2018*), or even materials with superhydrophobic textures thar are able to induce a delayed adhesion and inhibit the production of bacterial biofilm (*Falde et al., 2016*). The inhibition of the initial biofilm fixation by biomaterials is also a useful strategy for controlling biofilm formation. The biogenic AgNPs in the present study proved to be efficient not only in preventing bacterial growth, but significantly reduced biofilm formation in all *C. pseudotuberculosis* isolates; this situation is very pertinent, since the formation of biofilm by this specific bacterium is usually correlated with its virulence and with its capacity to induce a chronic disease (*Dorella et al., 2006*).

The initial attachment of cells to biofilm-forming surfaces occurs within the first two days of biofilm formation (*Subhadra et al., 2018*). AgNPs acted for 48 h in *C. pseudotuberculosis*, controlling the change between planktonic and biofilm-associated forms. This control was also observed in Gram-positive cocci treated with small molecules known as aryl rhodanines (*Opperman et al., 2009*), and in *Vibrio cholerae* treated with di-Cyclic GMP inhibitors (*Connera et al., 2017*). *Saising et al. (2012)* found that galidermin was able to completely inhibit biofilm formation at concentrations of 4 to 8 µg/mL; these authors also describe that the levels of transcription of the *atl* genes, which encode molecules that participates in the primary adhesion, and *ica* genes, which encodes an adhesin related to exopolysaccharide production, were significantly reduced in the presence of the antimicrobial agent. Antimicrobial peptides are also known to interfere with biofilm formation by different bacterial pathogens. Antibiofilm peptide 1018 blocked ppGpp, which is known to play an important role in biofilm formation, and the treatment with this peptide completely prevented biofilm formation and led to the eradication of mature biofilms formed by *P. aeruginosa*, *E. coli*, *Acinetobacter baumannii*, *K. pneumoniae*, methicillin resistant *S. aureus*, *Salmonella* Typhimurium and *Burkholderia cenocepacia* (*Fuente-Núñez et al., 2014*).

As a confirmation of the biofilm formation using a 96-well culture plate, we examined the formed biofilms by scanning electron microscopy. According to our results, the *C. pseudotuberculosis* strains demonstrated an ability to form mature biofilms. These results are in accordance with a previous study that observed the biofilm formation and consolidation by the *C. pseudotuberculosis* CAPJ4 strain (*De Sá et al., 2021*). We were able to observe that AgNP-treated *C. pseudotuberculosis* biofilms showed severe cell destruction and biofilm disruption. As described by other authors, the bacterial exposure to other types of AgNPs resulted in the adhesion of nanoparticles on the cell wall and membrane, causing significant morphological changes, such as cytoplasm shrinkage, membrane displacement and cell disruption (*Dakal et al., 2016*). SEM images of *P. aeruginosa* biofilm showed few bacterial aggregates and fewer viable bacteria after 24 h of exposure to nanoparticles (*Mu et al., 2016*). The 3 h treatment of *S. aureus* biofilm with catechin-copper nanoparticles dramatically changed the shape and size of the cells, resulting in wrinkled cell walls and

in the adhesion of many materials on the bacterial surface (*Li et al., 2015*), a result very similar to what was found in our images. The formation of vesicles on the cell surface of *C. pseudotuberculosis* is consistent with an increased cytoplasmic membrane permeability and leakage of cellular compounds (*Diao et al., 2004*). A similar result was found after exposure of *P. aeruginosa* to ozone (*Zhang et al., 2011*). Increased cytoplasmic membrane permeability and destabilization of cell structures and biomolecules are common AgNP and ozone mechanisms of action (*Habash et al., 2014*; *Singh et al., 2015*; *Dakal et al., 2016*).

## CONCLUSIONS

The *Fusarium oxysporum*-based biogenic AgNPs showed to be significantly effective against *C. pseudotuberculosis* in the planktonic form. Our study is the first to describe the action of nanoparticles on *C. pseudotuberculosis* biofilms and our findings indicate a high efficacy regarding the ability to cause biofilm disruption and destruction, with significant bacterial morphological changes. The use of biologically synthesized AgNPs represented an adequate strategy for biofilm formation interference. In this way, biogenic silver nanoparticles can be seen as promising antimicrobial agents for the control of caseous lymphadenitis in small ruminants.

## ACKNOWLEDGEMENTS

The authors give thanks to the Electronic Microscopy Unit of the Instituto Gonçalo Moniz, (FIOCRUZ/Bahia) for the use of the scanning electron microscope, and to Francisca Soares (LABIMUNO ICS/UFBA) for technical assistance.

### Funding

This study was funded by the Fundação de Apoio à Pesquisa e Extensão (FAPEX), through continuous resources obtained by extension projects. Laerte Marlon Santos and Mauricio Alcantara Kalil are PhD fellows from the Coordenação de Aperfeiçoamento de Pessoal de Nível Superior (CAPES). Ricardo Wagner Portela is a Technical Development fellow from the Conselho Nacional de Desenvolvimento Científico (CNPQ—Proc. 310058/2022-8). The funders had no role in study design, data collection and analysis, decision to publish, or preparation of the manuscript.

### Grant Disclosures

The following grant information was disclosed by the authors:
Fundação de Apoio à Pesquisa e Extensão (FAPEX), through continuous resources obtained by extension projects.
Laerte Marlon Santos and Mauricio Alcantara Kalil are PhD fellows from the Coordenação de Aperfeiçoamento de Pessoal de Nível Superior (CAPES).
Ricardo Wagner Portela is a Technical Development fellow from the Conselho Nacional de Desenvolvimento Científico: CNPQ—Proc. 310058/2022-8.

## Competing Interests

Vasco Azevedo and Debmalya Barh are Academic Editors for PeerJ.

## Author Contributions

- Laerte Marlon Santos conceived and designed the experiments, performed the experiments, prepared figures and/or tables, and approved the final draft.
- Daniela Méria Rodrigues performed the experiments, prepared figures and/or tables, and approved the final draft.
- Bianca Vilas Boas Alves performed the experiments, prepared figures and/or tables, and approved the final draft.
- Mauricio Alcântara Kalil performed the experiments, prepared figures and/or tables, and approved the final draft.
- Vasco Azevedo analyzed the data, authored or reviewed drafts of the article, and approved the final draft.
- Debmalya Barh analyzed the data, authored or reviewed drafts of the article, and approved the final draft.
- Roberto Meyer conceived and designed the experiments, analyzed the data, authored or reviewed drafts of the article, and approved the final draft.
- Nelson Duran conceived and designed the experiments, analyzed the data, authored or reviewed drafts of the article, and approved the final draft.
- Ljubica Tasic conceived and designed the experiments, authored or reviewed drafts of the article, and approved the final draft.
- Ricardo Wagner Portela conceived and designed the experiments, prepared figures and/or tables, authored or reviewed drafts of the article, and approved the final draft.

## Data Availability

All the raw data is available in the Supplemental Files.

## Supplemental Information

Supplemental information for this article can be found online at http://dx.doi.org/10.7717/peerj.16751#supplemental-information.

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
