# Peer review of "Activity of biogenic silver nanoparticles in planktonic and biofilm-associated Corynebacterium pseudotuberculosis"

_PeerJ, doi:10.7717/peerj.16751_

## Round 0.1 · original submission · Major Revisions

The reviewers were happy that the manuscript was of interest and for the most part well-executed/designed. However each reviewer did have concerns about the manuscript that will require addressing, in particular the written English.

**Language Note:** The Academic Editor has identified that the English language must be improved. PeerJ can provide language editing services - please contact us at copyediting@peerj.com for pricing (be sure to provide your manuscript number and title). Alternatively, you should make your own arrangements to improve the language quality and provide details in your response letter. – PeerJ Staff

·

Basic reporting

The subject approached in this manuscript is interesting, but the results is confusing and contradictory.
Some figures must be joined to better read the text.

Experimental design

The biggest question is about the choice of concentrations to evaluate the interference with biofilm formation. It was not possible to understand why concentrations above bactericidal concentrations were used to evaluate the interference with biofilm formation. According to supplementary figure 1 and 2, above 0.3 mg/mL there is 100% inhibition of bacterial growth

Validity of the findings

I thank you for providing raw data, however your supplemental files need translation conference. Some spreadsheets are in "Portuguese".

Reviewer 2 ·

Basic reporting

The abstract must be contain numerical results

Experimental design

Authors made biogenic nanoparticles with fungi
1-the methods must be clear
2-all characterization of nanoparticles must be added to material and methods and also in the results
3-in SEM image which concentrations were used to affect biofilm and make images
4- the authors mentioned the isolates are molecular identify, please add the accession number of isolates

Validity of the findings

The conclusion is well-written

Reviewer 3 ·

Basic reporting

Review for Santos et al “ACTIVITY OF BIOGENIC SILVER NANOPARTICLES IN PLANKTONIC AND BIOFILM-ASSOCIATED Corynebacterium pseudotuberculosis”

Santos et al have worked on the antimicrobial and antibiofilm activity of AgNPs on Corynebacterium pseudotuberculosis. They have determined the MIC and MBC of C. pseudotuberculosis. They have also done scanning electron microscopy to confirm the inhibitory effect of AgNPs on C. pseudotuberculosis biofilm. The experiments are thoroughly executed. However, the manuscript needs major language editing as it was not clear to me what the authors were trying to convey at various places.

My concerns-
Line 40 “It was found differences in the capacity of biofilm formation and response after exposure to AgNPs between reference strains and clinical isolates of C. pseudotuberculosis”
to
“AgNP significantly reduced the formation of biofilm in reference strains and clinical isolates of C. pseudotuberculosis”

Line 43 The term sessile is confusing. Does it mean biofilm? as biofilms are non-motile. change it to biofilm

Line 53 lymphadenitis (CL) the abbreviation is incomplete.

Line 58 remove the double hyphen.

Line 69 Currently, it is needed to Currently, there is a need

Line 197 and 198. The T1 strain is mentioned as both poor and moderate
biofilm producer. which category does it belong to?

Line 272 Give full forms of abbreviations MRSA, VRE, ESBL.

Line 274 Furthermore, these results proved the antimicrobial potential of
AgNPs against resistant strains isolated from human hospitals.
provide reference for this statement.
Line 279 What do you mean by consolidated biofilms?
Line 279 Please reframe the sentence. “In the analyzes……”
it does not make any sense what I am reading.

Figure 1 Use different contrasting colors in the bar graphs. strain N1 and
Isolate 06 has the same color bar

Experimental design

'no comment'

Validity of the findings

'no comment'

Additional comments

Review for Santos et al “ACTIVITY OF BIOGENIC SILVER NANOPARTICLES IN PLANKTONIC AND BIOFILM-ASSOCIATED Corynebacterium pseudotuberculosis”

Santos et al have worked on the antimicrobial and antibiofilm activity of AgNPs on Corynebacterium pseudotuberculosis. They have determined the MIC and MBC of C. pseudotuberculosis. They have also done scanning electron microscopy to confirm the inhibitory effect of AgNPs on C. pseudotuberculosis biofilm. The experiments are thoroughly executed. However, the manuscript needs major language editing as it was not clear to me what the authors were trying to convey at various places.

---

## Round 0.2 · accepted · Accept

Two of the three reviewers were happy that their previous concerns had been addressed. While reviewer 1 was less enthusiastic, I can see from the authors' rebuttal letter that there has been an effort to address this main concern, and that this has been explained in the manuscript.

The manuscript now appears ready to publish. Congratulations.

·

Basic reporting

The subject of the manuscript is interesting, since there is no research on silver nanoparticles in C. pseudotuberculosis. However, the choice of concentrations for biofilm treatment was not convincing. The authors selected concentrations above bactericidal, which are concentrations that eliminate bacterial growth.

Experimental design

No comment

Validity of the findings

No comment

Additional comments

No comment

Reviewer 2 ·

Basic reporting

We accept in present form to publish in the present form

Experimental design

we accept in present form

Validity of the findings

no comments

Additional comments

Accept

Reviewer 3 ·

Basic reporting

I commend the authors for their thorough investigation into the potential use of silver nanoparticles (AgNPs) against C. pseudotuberculosis. The study is well-structured, and the research addresses a significant gap in our understanding of the biofilm characteristics of C. pseudotuberculosis and its response to antimicrobial agents.

I appreciate the authors for their prompt and thorough revisions addressing my earlier concerns. The modifications have significantly strengthened the manuscript, and I am now satisfied with the completeness and clarity of the content.

Experimental design

no comment

Validity of the findings

no comment

Additional comments

I appreciate the authors for their prompt and thorough revisions addressing my earlier concerns. The modifications have significantly strengthened the manuscript, and I am now satisfied with the completeness and clarity of the content.